# Evolutionary Distributed Training

## Abstract

We introduce Evolutionary Distributed Training (EDT), a nature-inspired approach to distributed model training. EDT replaces centralized gradient synchronization with evaluation, pairwise model crossover, and mutation, enabling communication-efficient training across loosely connected devices. While early investigations show limited effectiveness in language model pretraining, EDT demonstrates strong potential in reinforcement learning (RL). In complex multi-agent environments, EDT facilitates diverse reward exploration and emergent strategies by evolving both policy and reward functions, outperforming traditional training in adaptability and strategic diversity. We also hypothesize EDT as a promising framework for post-training and alignment, offering optimization towards multi-objective, non-differentiable goals. This work positions EDT as a scalable, evolutionary recipe for distributed learning, offering early insights into where it may best fit within the deep learning landscape.

## 1   Introduction

The last decade of machine learning has been defined by scale. Transformer-based language models have demonstrated remarkable emergent capabilities when scaled to billions or even trillions of parameters, prompting discussions of "Sparks of Artificial General Intelligence" [2]. Recent success with Reinforcement Learning via Verifiable Reward (RLVR) brings us a step closer to human-level general intelligence [18][22]. Towards that goal, we ponder a more fundamental question — How did intelligence come to be in the first place? A simple answer would be — natural evolution. From the collective mind of an ant colony to the all-powerful human brain, intelligence emerges from an evolutionary algorithm. In this work, we intend to investigate how evolutionary algorithms may fit into the current landscape of deep learning. More specifically, we make two observations about nature: 1. Natural evolution is highly parallelized and decentralized. 2. Most complex organisms in nature perform pairwise reproduction. From this, we propose Evolutionary Distributed Training (EDT), a nature-inspired recipe for applying evolutionary algorithms as an outer layer to distributed training. Models are evaluated locally, selected based on fitness, and recombined via pairwise crossover and mutation to produce offspring. This process repeats iteratively, without requiring centralized synchronization of gradients. In the following sections we will conduct some early exploration of EDT on Large Language Model (LLM) Pre-training, Post-training, and Reinforcement Learning (RL).

## 2   Language Model Pretraining

### 2.1   Motivation

Recent work on distributed language model training has introduced new ways to scale model training beyond traditional data-parallelism. In particular, DiLoCo [5], a variant of federated averaging [11]

presents a promising distributed optimization algorithm that matches the performance of data-parallel while communicating 500 times less [5]. DiLoCo works by periodically synchronizing model deltas with a central aggregator using SGD with momentum and Nesterov acceleration. Inspired by the biological process of pairwise reproduction, we propose a decentralized alternative to DiLoCo using our Evolutionary Distributed Training (EDT) recipe. Rather than averaging deltas across all workers, we perform pairwise averaging, producing offspring that inherit characteristics from two parent models. Our hope is that the momentum accumulated over an ancestry might act as a larger batch, while the evolutionary algorithm optimizes for better-performing ancestries that carry the right momentum.

## 2.2 Method and Experimental Setup

We initialize a 6-million parameter language model based on the LLaMA architecture [21] and GPT-2 tokenizer [15]. The training dataset is derived from TinyStories [6].

To adapt the DiLoCo algorithm, we replace global averaging with pairwise model crossover. The training loop proceeds as follows:

1. **Fitness Evaluation:** Each model is evaluated on a fixed and shared validation set consisting of **400 examples**.

2. **Selection:** We perform **rank-based selection**, where models with higher fitness scores are more likely to be selected as parents.

3. **Crossover:** For each pair of parent models $(A, B)$:
   - Compute the base model as a linear interpolation of their pre-update checkpoints.
   - Average their training deltas to form a gradient.
   - Average the velocity buffers.
   - Apply the gradient to the base model using SGD with momentum and Nesterov acceleration.
   - Carry over the resulting model to the next generation.

4. **Local Training:** Each worker performs **400 inner steps** using the **AdamW optimizer**.

We compare the following configurations:

1. **DiLoCo Baseline:** Running on **8 workers** with a per-device **batch size of 1**.

2. **EDT with Rank-Based Pairwise Selection:** Running on **8 workers** with a per-device **batch size of 1**.

3. **EDT with Random Pairwise Selection:** Running on **8 workers** with a per-device **batch size of 1**.

4. **EDT with Increased Update Frequency:** Using $4\times$ more frequent updates (100 inner steps) on **8 workers** with a per-device **batch size of 1**.

## 2.3 Results

Figure 1 presents the validation perplexity across 32 outer steps. We observe that DiLoCo outperforms EDT significantly over the course of training. While DiLoCo baseline shows signs of continuing to decline in perplexity, our pairwise method appears to be slowing down to a stop. Furthermore, rank-based and random pairwise selection yield nearly identical learning curves, indicating that the evolutionary selection mechanism does not significantly influence model learning in this setting.

Increasing the update frequency in EDT (4x the updates) leads to faster initial perplexity reduction. However, all configurations converge to similar perplexity values after equivalent total training steps. This suggests that more frequent mixing may improve short-term convergence but does not overcome EDT's inherent limitations in aggregating knowledge or carrying gradient information across generations.

Given initial exploration results, we conclude that EDT does not seem to fit directly in pre-training settings.

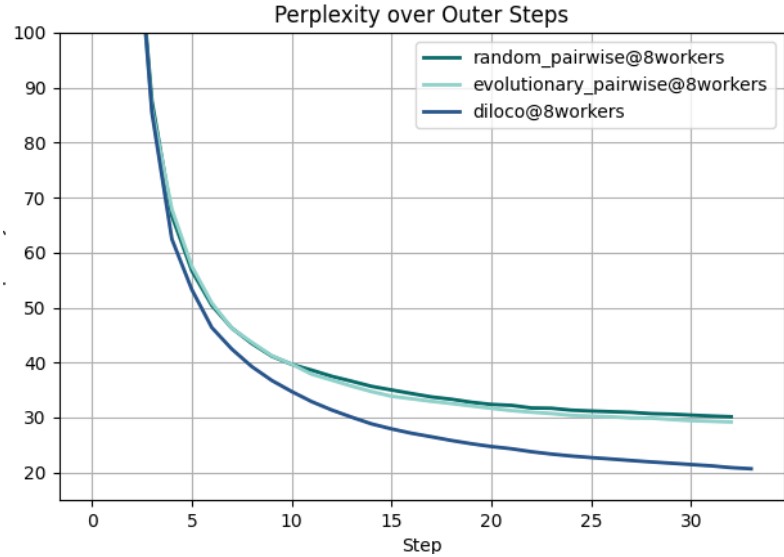

Figure 1: Comparing perplexity over training steps for random selection, evolutionary selection and DiLoCo baseline

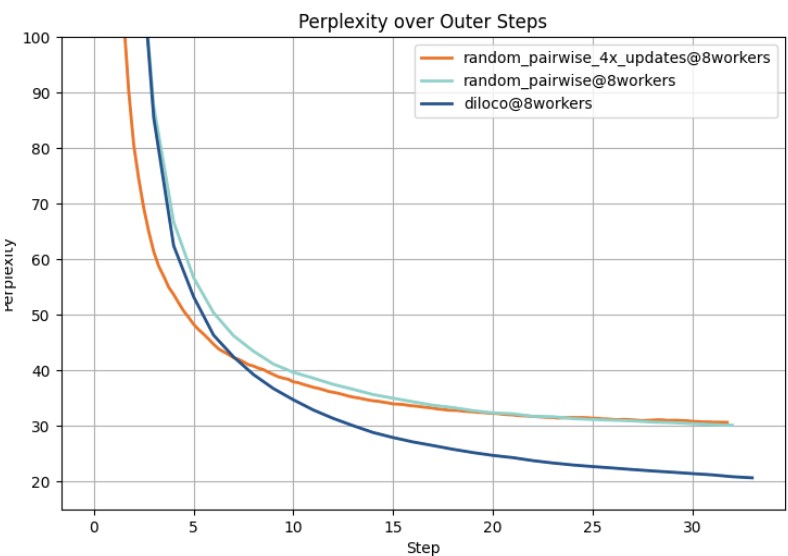

Figure 2: 4x updates shows no improvement in overall perplexity score

# 3 Reinforcement Learning

## 3.1 Motivation

Reinforcement Learning (RL) has achieved remarkable success across domains, but remains funda-mentally constrained by its reliance on well-shaped reward functions. Sparse or deceptive reward landscapes often result in agents that fail to learn meaningful behaviors. If you use a simple reward function for a complex or long-horizon problem, the model likely will have a hard time learning anything meaningful due to having too much space to explore. While if you manually create a complex reward function, you risk the model overfitting and gaming the reward. In the end we are stuck in an exploration-exploitation dilemma. One solution to this is by scaling up batch size with massively parallel environments [1]. Such a method requires a substantial number of rollout workers [13], which poses significant infrastructural challenges.

We propose that **Evolutionary Distributed Training (EDT)** offers a natural mechanism to encourage exploration without requiring handcrafted rewards or dense feedback.

To this end, we extend EDT to reinforcement learning by treating **combinations of reward functions as mutable DNA sequences**. Each worker trains a local policy model using its own reward configuration, allowing a population of agents to explore diverse behaviors. Models are selected based on performance (e.g., Elo rating in self-play), and offspring inherit both policy parameters and modified reward functions through crossover and mutation. This introduces structured randomness into the optimization landscape, promoting a balance between exploitation of high-performing strategies and exploration of novel ones.

Unlike the LLM pretraining setting, where averaging gradients is key, RL is inherently noisy and local. The decentralized and diverse nature of EDT aligns naturally with RL's challenges.

Furthermore, consider the following **analogy**: Say you want to find a treasure. You decided to make a thousand clones of yourself to do the searching. However, since these are your clones, they behave similarly, they all have the same habits, same ways of thinking. Maybe you will be able to find the treasure one day, but in the process you are doing a lot of repeated exploration, looking under the same rock multiple times. Your search space is confined by your function space [12]. What if instead of a thousand you, we have a thousand different people, each with their own quirks and behaviors, we expand the search space – EDT could promote a more efficient random search for solutions.

## 3.2 Method and Experimental Setup

We experiment with EDT-RL in a Smash Bros style platform fighting game gym environment [10]. Agents control characters that can move, jump, and perform light or heavy attacks in 8 directions, etc, with the objective of knocking their opponent off the platform.

**Model:** Each agent is a 64k-parameter transformer-based policy model, trained via Proximal Policy Optimization (PPO) [17].

**Fitness:** We use an Elo rating system based on double-elimination tournaments to evaluate model performance via self-play matches within the population.

**Reward DNA:** There are 18 different reward functions, including noop. Each agent is assigned a "DNA sequence" of size 6 encoding the combination of reward functions.

**Training Procedure:**

1. Each worker trains a local policy guided by its reward DNA for one generation.
2. After evaluation, agents are selected based on Elo score (fitness).
3. Pairs of parents are chosen to produce offspring:
   - **Policy crossover**: Spherical linear Interpolate weights of parent models.
   - **DNA crossover**: Uniformly mix reward shaping terms from both parents.
4. A mutation step applies to a subset of the offspring, randomly altering components of the DNA (e.g., replacing and introducing new reward functions).

We compare EDT-RL to a standard Population-Based Training (PBT) setup with fixed, manually defined reward shapings and identical PPO hyperparameters.

## 3.3 Results

In the 8-worker setting, EDT-RL produces competitive performance relative to PBT.

Scaling to **32 workers over 40 generations** reveals striking emergent behavior. Certain reward functions, while harmful in isolation, led to novel strategies when combined and refined across generations. One such mutation was the `Floor is Lava` penalty, which forces the model to keep jumping. Another penalized the use of light attacks. While suboptimal at first, these mutations encouraged exploration of aerial tactics. By generation 20, a new strategy emerged—**"ground slamming"** — where an agent jumped above their opponent and executed heavy vertical attacks repeatedly. This tactic gained dominance after agents discovered that stacking enough damage enabled a physics engine exploit: slamming opponents through the stage floor, bypassing typical

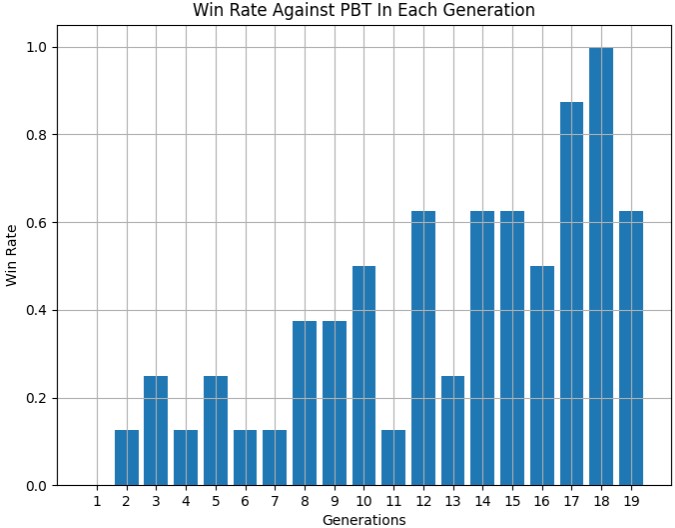

Figure 3: EDT@8workers vs PBT@8workers win rate at each generation. (Tie is considered a loss.)

knockback mechanics. By generation 40, this strategy dominates the population. However, the "floor is lava" gene that led to this strategy became scarce over the generations as agents that are not confined by this restriction were able to combine grounded strategies with "ground slam", achieving greater performance.

## 3.4 Discussion

These results highlight the unique strengths of EDT-RL:

- **Exploration via Mutation:** Introducing stochastic reward variation enabled the population to discover creative, high-impact strategies that were not explicitly encoded in the reward design.
- **Pairwise Crossover:** Pairwise crossover encourages diversity while maintaining the ability to propagate knowledge across the population, allowing efficient use of compute.
- **Scalability:** Larger populations enabled deeper exploration and more effective recombination, supporting rapid behavioral innovation. O(1) communication overhead allows decentralized and potentially asynchronous training over poorly connected devices.

# 4 Language Model Post-Training - Concept

## 4.1 Motivation

While pretraining focuses on fitting to large-scale data distributions, post-training aims to steer pretrained language models toward helpful, safe, and capable behavior—often using only a small amount of data. At this stage, we are not optimizing for maximum likelihood, but for alignment with specific goals and constraints. This makes post-training a natural candidate for **Evolutionary Distributed Training (EDT)**.

Post-training often involves multi-objective tradeoffs (e.g., helpfulness vs. harmlessness), non-differentiable goals (e.g., model judged quality), and the risk of overfitting to narrow supervision signals. Evolutionary algorithms are well-suited for such settings due to their ability to optimize:

- **Multi-objective functions** without requiring scalar reward collapse.
- **Non-differentiable objectives**, such as evaluations from LLM-as-a-judge or task-specific win-rates.

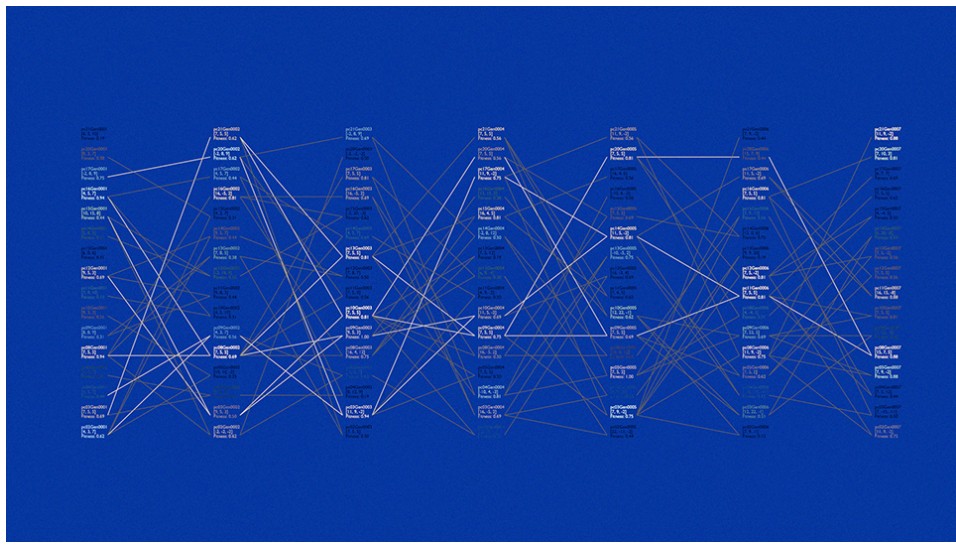

Figure 4: EDT post-training framework concept

- **Policy-space exploration**, encouraging behavioral diversity and robustness.

We propose EDT as a flexible wrapper around post-training. Models can be evaluated using any mix of automated metrics, benchmarks, or preference judgments. Pairwise crossover and mutation allow exploration across behavioral variants, while decentralized evaluation prevents synchronization bottlenecks. The goal is not to fine-tune for narrow targets, but to *stabilize training, prevent overfitting, and encourage emergent capabilities* through a process of natural evolution.

Due to computational constraints, we are able to conduct experiments at the moment. This direction could be explored in future work.

## 5 Related Work

### 5.1 Evolutionary Algorithms and Genetic Optimization

Evolutionary algorithms (EAs) have long been used to solve optimization problems through principles inspired by natural selection. Early work in this domain includes Genetic Algorithms (GAs) [7] which use crossover and mutation to evolve solutions over generations. In the context of neural networks, Neuroevolution techniques like NEAT (NeuroEvolution of Augmenting Topologies) [20] explore ways to evolve both architectures and weights. More recent large-scale efforts such as OpenAI's Evolution Strategies [16] demonstrated scalability of black-box optimization for deep reinforcement learning, rivaling traditional policy gradients in some tasks.

While these methods are powerful, they are rarely used in the era of large-scale language models due to their sample inefficiency and communication overhead. Our proposed Evolutionary Distributed Training (EDT) differs in that it applies EA concepts specifically to decentralized training settings — targeting communication-constrained environments and non-differentiable objectives.

### 5.2 Federated Learning and Distributed LLM Training

Federated learning [11] enables decentralized optimization by allowing clients to train local models and periodically synchronize updates with a central server. While originally proposed for privacy-preserving applications, its communication efficiency has inspired variants in large-scale language model training.

Recent work such as DiLoCo (Distributed Low-Communication) [5], shows extremely competitive performance in Large Language Model training compared to data-parallel, while supported by scaling laws [3] and open source reproduction [8].

## 5.3 Reinforcement Learning and Exploration

Exploration remains a central challenge in reinforcement learning, particularly in environments with sparse or deceptive rewards. Classical techniques include $\epsilon$-greedy exploration, entropy regularization, etc. More recently, large-scale self-play has proven effective in discovering complex strategies, as in AlphaZero [19] and OpenAI Five [13].

Evolutionary approaches have also been used in RL to promote exploration. Novelty Search [9] and Quality-Diversity algorithms [4] evolve agents to encourage behavioral diversity, often outperforming reward-based methods in hard-exploration domains. Population-Based Training (PBT) [?] introduced hybrid training-evolution schemes, combining gradient descent with hyperparameter evolution. Our EDT-RL builds on these ideas, extending the concept to evolve both policy weights and reward functions in a decentralized fashion.

## 5.4 Post-Training and Model Steering

Post-training — the phase following large-scale pretraining — is where language models are adapted to specific behaviors and goals. Typical methods include supervised fine-tuning on curated instructions [14], Reinforcement Learning from Human Feedback (RLHF) [? 14], and more recently, Reinforcement Learning via Verifiable Rewards (RLVR) [18].

## 6 Limitations

Most of the initial experiments were done on a single laptop RTX4070 by simulating having multiple workers in a loop. Later, we found a way to scale up to 32 RTX4080s, and was able to complete most experiments in the RL section. However, when we started experimenting with post-training, we were noticed by the university and informed that personal research is not allowed on university infrastructure. Due to limitation in compute, our experimental results are limited.

## 7 Conclusion

We introduced Evolutionary Distributed Training (EDT), a nature-inspired recipe for decentralized and parallelized model training grounded in the principles of natural evolution — local evaluation, pairwise reproduction, and mutation. While our initial experiments show that EDT does not offer significant advantages in language model pertaining, it shows strong promise in reinforcement learning. In particular, EDT-RL enabled the emergence of novel strategies through reward function mutation and policy recombination, demonstrating its potential as a tool for structured exploration in sparse or deceptive environments.

This work represents an early investigation into the possible roles evolutionary algorithms can play in distributed training systems. We intend to use this paper as a stepping stone towards collecting feedback and finding potential collaborators.

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
