# OpenReview forum: "Evolutionary Distributed Training"
_NeurIPS.cc/2025/Conference — Submitted to NeurIPS 2025_

### Official Review · Reviewer_b2TS · 2025-06-26

**Clarity:** 2
**Significance:** 1
**Originality:** 1
**Rating:** 2
**Confidence:** 4

**Summary:**

The authors propose an evolutionary distributed training method that replaces centralized gradient synchronization with evaluation, paired model cross-over, and mutation, enabling efficient training across loosely connected devices.

**Questions:**

Please refer to Strengths and Weaknesses.

**Ethical Concerns:**

["NO or VERY MINOR ethics concerns only"]

**Final Justification:**

The author did no rebuttal, and my score will not change.

**Limitations:**

Please refer to Strengths and Weaknesses.

**Paper Formatting Concerns:**

No formatting issues

**Quality:**

2

**Strengths And Weaknesses:**

The paper is novel and interesting, but there are still some shortcomings.
1) Although the proposed evolutionary distribution training is interesting, it lacks relevant theoretical support.
2) The structure and layout of the paper are hard to read.
3) There are limitations in the experimental design and results, including insufficient comparative experiments, insufficient depth of result analysis, and a lack of specific application scenarios.
4) To better validate the effectiveness of the proposed evolutionary distribution training, the paper should include experiments on larger-scale benchmark datasets, such as ImageNet, CIFAR-100, and COCO. The lack of such evaluations represents a major limitation of this study.
5) The overall structure of this paper requires substantial revision. The current organization suffers from issues related to information flow and writing clarity, which hinder the effective communication of the paper’s core contributions.

---

> ### Comment · Reviewer_b2TS · 2025-08-05
> **No rebuttal**
>
> No rebuttal, I keep my score.

---

### Official Review · Reviewer_5HDw · 2025-07-02

**Clarity:** 1
**Significance:** 1
**Originality:** 1
**Rating:** 1
**Confidence:** 5

**Summary:**

This paper attempts to propose an evolutionary approach as a way to perform distributed training of models, instead of centralized gradient synchronization. Local models are tested on some fixed validation set or validation episodes and are updated using an evolutionary algorithm. The paper performs preliminary experiments on a language pre-training task, a two player reinforcement learning task, and a language post-training task.

**Questions:**

I have no questions.

**Ethical Concerns:**

["NO or VERY MINOR ethics concerns only"]

**Final Justification:**

My recommended score is 1. This paper has an incomplete related work section and preliminary section. Moreover, the experiments are also not explained in detail, a some plots seem incomprehensible.

**Limitations:**

yes

**Paper Formatting Concerns:**

The paper seems to be formatted aptly.

**Quality:**

1

**Strengths And Weaknesses:**

Weaknesses:
1) Unfortunately, I think the current paper is very preliminary, missing discussion on background (no discussion on what DiLoCo is and how it relates to the proposed method) and missing definition of used technical terms (what is model delta, velocity buffer, spherical linear interpolate? What are the 18 different reward functions? What is a standard Population-Based Training setup?).
2) There are no results for the post-training experiments. Figure 4 is incoherent, and no explanation for it is provided.
3) There are no error bars in any of the plots.

---

> ### Comment · Reviewer_5HDw · 2025-08-05
> **Reply by reviewer**
>
> No rebuttal, I keep my score.

---

### Official Review · Reviewer_XkEj · 2025-07-02

**Clarity:** 2
**Significance:** 1
**Originality:** 2
**Rating:** 2
**Confidence:** 4

**Summary:**

This paper introduces Evolutionary Distributed Training (EDT), a nature-inspired approach to distributed model training. The idea is built over the DiLoCo paper, but it replaces centralized gradient synchronization with evaluation, pairwise model crossover, and mutation. It proposes local evaluation , pairwise model crossover via merging , and model mutation with reward combination. The paper is in a very preliminary stage, as they performed initial experiments on pre-training and reinforcement learning (RL).
Their initial results show that pre-training do not benefit from the approach, but RL does.

**Questions:**

- Validation set for pre-training is too small 400 examples
- Linear interpolation was used to merge models, do you try other merge approaches?
- Could you make a time/complexity analysis of the problem, exploration could be expensive.

**Ethical Concerns:**

["NO or VERY MINOR ethics concerns only"]

**Limitations:**

- The authors honestly stated that this work is preliminary and incomplete

**Quality:**

3

**Strengths And Weaknesses:**

- The paper demonstrates a use case for EDT in reinforcement learning (RL). By evolving both policy and reward functions, EDT-RL successfully fostered diverse exploration and led to the emergence of  effective strategies (e.g., "ground slamming") that were not explicitly designed. However, there is large literature for applying evolutionary algorithms for agents in game environments.
- Paper is incomplete.
- The experiments in both pre-training and RL are done with very small models. Even preliminary experiments should be done in larger scale.
- Missing literature. Please check this paper and their related work: https://arxiv.org/pdf/2410.11163

---

### Official Review · Reviewer_dhmT · 2025-07-18

**Clarity:** 2
**Significance:** 1
**Originality:** 1
**Rating:** 2
**Confidence:** 3

**Summary:**

The authors propose a method for scaling a standard genetic algorithm to distributed training. The method struggles to benefit LLM pretraining, but shows some promise in RL settings. The authors compare their approach to federated learning methods that split their data across multiple machines and collaboratively train large models. In the conclusion, the authors note this is an exploratory work that they hope will yield feedback and potential collaborations. I’ll focus my feedback to help guide the authors to improving this work.

**Questions:**

- A key point of the paper is that EDT does not require "centralized synchronization of gradients." The implication of this seems to be that EDT would be faster than an algorithm like DiLoCo. Is there any metric that can support this, such as overall runtime, or the amount of time spent communicating gradients? It may also be helpful to include how much "gradient communication" is expected for each method, e.g., using big-O notation.

- It’s not clear to me how the proposed method fits in with other projects that have scaled evolutionary algorithms to distributed settings. I point the authors to the following papers (https://dl.acm.org/doi/abs/10.1145/3321707.3321817, https://arxiv.org/abs/1703.03864, https://arxiv.org/abs/2202.01258). Are these reasonable baselines that could be added? Given the strong performance on RL domains, it seems reasonable to add evolutionary RL baselines that were also distributed.

**Ethical Concerns:**

["NO or VERY MINOR ethics concerns only"]

**Limitations:**

yes

**Paper Formatting Concerns:**

Line 46: 6 million -> 6 billion?
Line 84-92, 104-110: Avoid use of second person ("you")

**Quality:**

2

**Strengths And Weaknesses:**

- I think comparing against federated learning is a good idea. I’m not familiar with another work that has compared evolutionary algorithms and federated learning. DiLoCo seems like a reasonable baseline for this comparison.
- The paper appears to be rather broad in the number of problems it is tackling. The three problems mentioned (pre-training, RL, and post-training) are all uniquely difficult problems in their own right; each one may even warrant an individual paper. I suggest selecting a single problem and focusing on that one.
- The mechanisms used in the paper could be much better motivated. For example, what motivates the crossover mechanisms on lines 54-59? Similarly, what motivates the training procedure on lines 121-128? Furthermore, is there any (empirical) evidence to support that such mechanisms perform as expected? I think comparing the random vs. evolutionary selection in Section 2 is a good step in this direction.

- For experimental results, it would be nice to eventually see a table listing the results (this is easier than trying to determine exact results from plots). Ideally, it would also be good to perform multiple runs of each algorithm and compare the results using statistical tests (such as t-tests and Wilcoxon tests).

- Regarding compute limitations, I would recommend reaching out to faculty at the university to see if they are willing to collaborate on this project.

---

> ### Comment · Reviewer_dhmT · 2025-08-04
>
> There appears to be no rebuttal.

---

### Decision · Program_Chairs · 2025-09-17

**Decision:**

Reject

**Comment:**

Reviewers all find the work premature with lack of clarity, incomplete or poor experiments, missing baselines from the evolutionary RL literature, and attempting to cover too many problem areas without depth. Evolutionary Distributed Training (EDT) replaces centralized gradient synchronization with evolutionary mechanisms such as crossover, mutation, and local evaluation. The idea is positioned as a communication-efficient alternative to federated learning. Results show little benefit for pretraining, some promise in RL, and no conclusive results elsewhere. There is no rebuttal and scores lean to strong reject.